# COVID-19 vaccine effectiveness among healthcare workers during the Omicron period in the country of Georgia, January – June 2022

Caleb L. Ward[1*◐], Madelyn Yiseth Rojas Castro[2◐], Giorgi Chakhunashvili[3], Nazibrola Chitadze[3], Iris Finci[4], Richard Pebody[4], Esther Kissling[2], Mark A. Katz[4], Lia Sanodze[3]

1 Public Health Institute, Tbilisi, Georgia, 2 Epiconcept, Paris, France, 3 National Center for Disease Control and Public Health, Tbilisi, Georgia, 4 World Health Organization Regional Office for Europe, Copenhagen, Denmark

◐ These authors contributed equally to this work.
* calebward94@gmail.com

## Abstract

### Introduction

Understanding COVID-19 vaccine effectiveness (VE) in healthcare workers (HCWs) is critical to inform vaccination policies. We measured COVID-19 VE against laboratory-confirmed symptomatic infection in HCWs in the country of Georgia from January – June 2022, during a period of Omicron circulation.

### Methods

We conducted a cohort study of HCWs in six hospitals in Georgia. HCWs were enrolled in early 2021. Participants completed weekly symptom questionnaires. Symptomatic HCWs were tested by RT-PCR and/or rapid antigen test (RAT). Partic[ipants were also routinely tested, at varying frequencies during the study period, for SARS-CoV-2 by RT-PCR or RAT, regardless of symptoms. Serology was collected quarterly throughout the study and tested by electrochemiluminescence immunoassay for SARS-CoV-2 antibodies. We estimated absolute and relative VE of a first booster dose compared to a primary vaccine series as (1-hazard ratio)*100 using Cox proportional hazards models.

### Results

Among 1253 HCWs, 141 (11%) received a primary vaccine series (PVS) and a first booster, 855 (68%) received PVS only, and 248 (20%) were unvaccinated. Most boosters were BNT162b2 (Comirnaty original monovalent) vaccine (90%) and BBIBP-CorV vaccine (Sinopharm) (9%). Most PVS were BNT162b2 vaccine (68%) and BBIBP-CorV vaccine (24%). Absolute VE for a first booster was 40% (95%

**Data availability statement:** The data has been restricted according to the description of data sharing outlined in the informed consent form of the study. This informed consent form, along with the protocol, was approved by the WHO and Georgia NCDC ethical review committees; these committees both require study investigators to adhere to data sharing agreements described in the protocol and the informed consent forms. The Institutional Review Board of the National Center for Disease Control and Public Health of Georgia can contacted for data requests. Their website is www.ncdc.ge.

**Funding:** This study was supported by funding from the World Health Organization/European Regional Office, the US Centers for Disease Control and Prevention (Cooperative Agreement No.: NU2GGH002093), the Task Force for Global Health and the Public Health Institute.

**Competing interests:** The authors have declared that no competing interests exist.

Confidence Interval (CI) -56–77) at 7–29 days following vaccination, -9% (95% CI -104–42) at 30–59 days following vaccination, and -46% (95% CI -156–17) at ≥60 days following vaccination. Relative VE of first booster dose compared to PVS was 58% (95% CI 1–82) at 7–29 days following vaccination, 21% (95% CI -33–54) at 30–59 days following vaccination, and -9% (95% CI -82–34) at ≥60 days following vaccination.

## Conclusion

In Georgia, first booster dose VE against symptomatic SARS-CoV-2 infection among HCWs was moderately effective but waned very quickly during Omicron. Increased efforts to vaccinate priority groups in Georgia, such as healthcare workers, prior to periods of anticipated high COVID-19 incidence are essential.

## Introduction

COVID-19 has caused high rates of illness and mortality in health care workers (HCWs) [1]. HCWs are at high risk of occupational exposure, and ensuring the safety and health of HCWs is critical for maintaining functioning healthcare systems in pandemic and interpandemic periods.

COVID-19 vaccines are an important tool to reduce morbidity and mortality from SARS-CoV-2 infections [2,3]. Reducing SARS-CoV-2 infections among HCWs can reduce work absences and also reduce the possibility of onward transmission to vulnerable patients [4,5]. While COVID-19 vaccine effectiveness (VE) has been widely studied in high-income countries (HICs), few studies have evaluated COVID-19 booster dose VE in middle-income countries (MICs) in Europe, where populations are different, and a variety of COVID-19 vaccines have been used in resource-limited settings.

Georgia is an upper-MIC in the South Caucasus with a population of 3.7 million people [6]. Four two-dose COVID-19 vaccines are currently approved for use among adults in the Republic of Georgia: Oxford/AstraZeneca AZD1222, Pfizer/BioNTech BNT162b2, Sinopharm (Beijing) BBIBP-CorV, and Sinovac (CoronaVac) [7]. HCWs were a priority group for vaccination after vaccines became available on 15 March 2021 [8]. As of May 2023, in Georgia, 34.6% of the population had received at least one dose, 31.6% had received a PVS, and 6.7% had received a booster dose [9].

In April 2021, we initiated a prospective cohort study of COVID-19 VE in HCWs in Georgia. VE estimates against the Delta variant from this study have been previously published [10], but VE estimates against the Omicron variant have not been previously published. Georgia was heavily affected by the Omicron wave; daily new cases peaked in early February, 2022 [11]. Here we evaluated COVID VE during the early Omicron period using data from the same cohort (January 1 – June 1, 2022).

## Materials and methods

This cohort study has been described previously [10,12,13]. Briefly, we enrolled HCWs from six hospitals in Georgia. We collected information on

socio-demographics, underlying conditions, hospital role, self-assessed health status, prior SARS-CoV-2 infection, and COVID-19 history. Every participant provided a blood sample for serology testing at the time of enrolment, and quarterly during the course of the study. Serology samples were tested for anti-nucleocapsid antibody (Anti-N) and anti-spike antibody (Anti-S), as previously described [10].

Throughout the study timeline, participants completed a weekly symptom questionnaire, administered by study personnel. Participants who reported any COVID-19 symptoms included in the Georgia Ministry of Health (MoH) suspected COVID-19 definition (fever, cough, general weakness, fatigue, headache, muscle aches, sore throat, runny nose, shortness of breath, lack of appetite, nausea, vomiting, diarrhea, altered mental status, loss of taste, or loss of smell) provided a respiratory specimen, which was tested for SARS-CoV-2 by PCR or RAT. During the entire study period, as part of the national COVID-19 response, HCWs at all six hospitals were recommended to be tested every week for SARS-CoV-2 by RT-PCR or RAT, but in practice testing was done inconsistently.

Participants who tested positive for SARS-CoV-2 by RT-PCR or RAT were administered a follow-up questionnaire about their clinical course 30 days after the date of their positive test. Data on testing and vaccination were confirmed using national databases.

## Vaccine effectiveness analysis

As our primary analysis, we estimated first booster dose VE against PCR or RAT confirmed symptomatic SARs-CoV-2 infection from January 1 to June 1, 2022, a period when >50% of weekly SARS-CoV-2 sequences in Georgia were Omicron variants BA.1 and BA.2, according to data from the Global Initiative on Sharing All Influenza Data (GISAID) [14] (Fig 1). Our secondary analysis estimated the relative VE of the first booster dose compared with primary vaccination series (PVS) among the population eligible for receive a first booster dose. We measured VE at discrete time intervals since receipt of booster dose: 7 days – 29 days, 30 days – 59 days, and ≥ 60 days. We also estimated VE specifically for BNT162b2 and against medically attended COVID-19.

Participants were considered to be fully vaccinated with PVS ≥ 14 days after their second COVID-19 vaccine dose, and for booster doses ≥7 days after the booster vaccination. In order to be consistent with Georgia MoH guidelines,

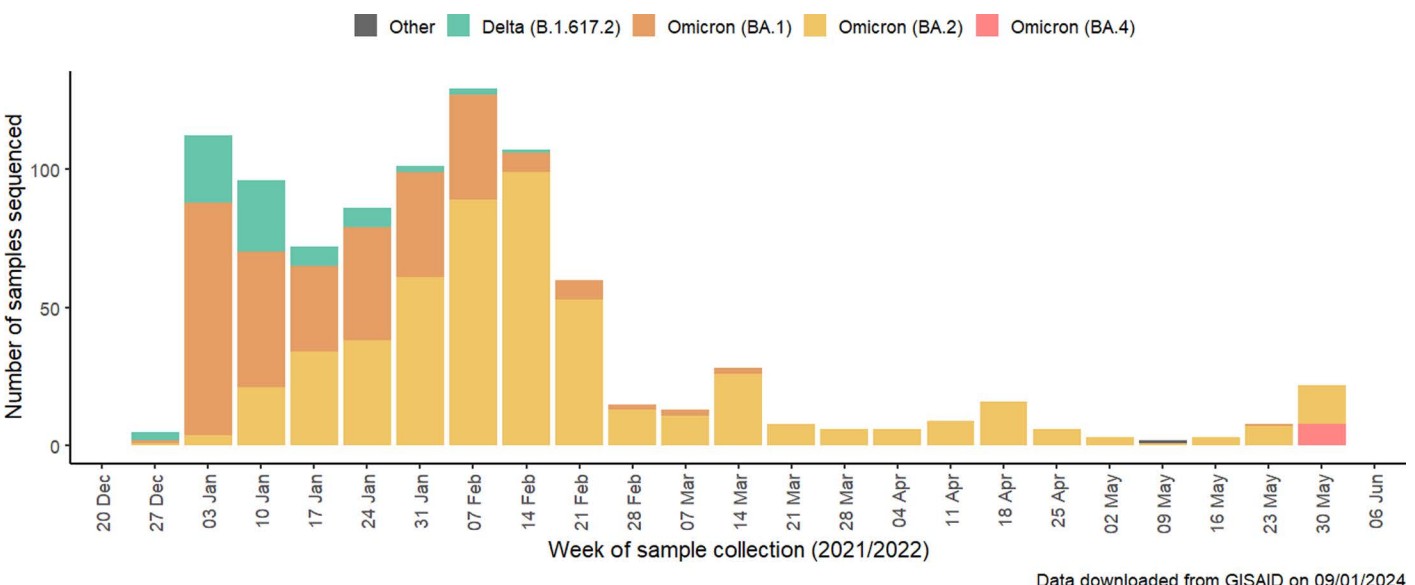

**Fig 1. SARS-CoV-2 variant distribution in Georgia, data from GISAID [ 14].**

participants were considered to be eligible for a first booster dose ≥150 days after PVS. If participants reported a previous PCR-confirmed SARS-CoV-2 infection they were considered as having a prior infection. In addition, unvaccinated participants and participants who had not received any inactivated vaccines were seropositive for Anti-Nucleocapsid antibodies at the beginning of the analysis period were considered to have previous infection. We could not use serology test results to determine prior infection for participants who were previously vaccinated with an inactivated vaccine (BBIBP-CorV or CoronaVac), as these vaccines routinely generate anti-nucleocapsid antibodies, and therefore positive serology test results in these participants could reflect either prior infection or prior vaccination.

## Statistical model

We estimated VE as (1 – hazard ratio)*100. Hazard ratios comparing vaccinated and referenced groups were estimated using Cox proportional hazards models with vaccination as a time-varying exposure. Vaccination status of some individuals changed over time from unvaccinated to PVS, and from PVS to boosted. Person-time from participants with either only one dose or more than three doses were excluded. Calendar time was used as the underlying time in the Cox regression. We calculated unadjusted and adjusted VE including the hospital site as strata to account for site-specific variation. We adjusted the multivariable regression model using a priori fixed covariates: age groups, sex, any underlying chronic condition (yes/no), self-assessed health status, household size, daily face-to-face patient contact, previous SARS-CoV-2 infection (confirmed by RT-PCR, RAT, or serology), smoking status. These variables were chosen because they were previously found to be associated with vaccine uptake and effectiveness [15–17].

Participants contributed person-time from the start of the study period (1 January 2022), or from 90 days after a prior SARS-CoV-2 infection (whichever was later), until whichever of the following endpoints came first: 1) the day of the first SARS-CoV-2 symptomatic infection 2) the day of receipt of a second booster vaccine dose, or 3) the day of the last weekly questionnaire before complete loss to follow-up, or withdrawal from the study.

The study was approved by the Georgia NCDC and WHO Research Ethics Review Committees (reference numbers IRB 2021–014 and CERC.0097B, respectively). The CDC humans review determined the activity to be a public health evaluation. All participants provided written informed consent. The study is also registered at clinicaltrials.gov (Identifier NCT04868448).

## Results

On January 1, 2022, there were 1592 HCWs enrolled in the study; 1253 were included in this analysis (Fig 2). Of those included in the analysis, the median age was 41 (Interquartile range: 30–53), 1058 (84%) were female, and 316 (25%) reported having at least one chronic condition. Most HCWs were nurses or midwives (39%) and medical doctors (21%). The majority of HCWs (64%) had evidence of prior SARS-CoV-2 infection at the beginning of the analysis period (Table 1).

At the beginning of the study period, 855 (68%) had received PVS, 141 (11%) had received a first booster dose, and 248 (20%) remained unvaccinated. Most boosters were BNT162b2 vaccine (92%) and BBIBP-CorV vaccine (8%), and most PVS were BNT162b2 vaccine (68%) and BBIBP-CorV vaccine (24%) (Table 1).

During the analysis period, there were a total of 372 PCR or RAT-confirmed SARS-CoV-2 symptomatic infections; 63 symptomatic infections among unvaccinated participants, 264 symptomatic infections among PVS, and 45 symptomatic infections among those who received a first booster dose (Fig 3). Overall, 267 (68%) of infected participants completed the clinical course at 30 days questionnaire. In total, 126 (47%) of infected participants sought medical care, and 8 (3%) were hospitalized.

Absolute booster dose VE against symptomatic SARS-CoV-2 infection was 40% (95% CI -56–77) in the 7–29 days following receipt of the first booster dose, -9% (95% CI -104–42) for 30–59 days following receipt of the first booster dose, and -46% (95% CI -156–17) for ≥ 60 days following receipt of the first booster dose (Table 2, Fig 4).

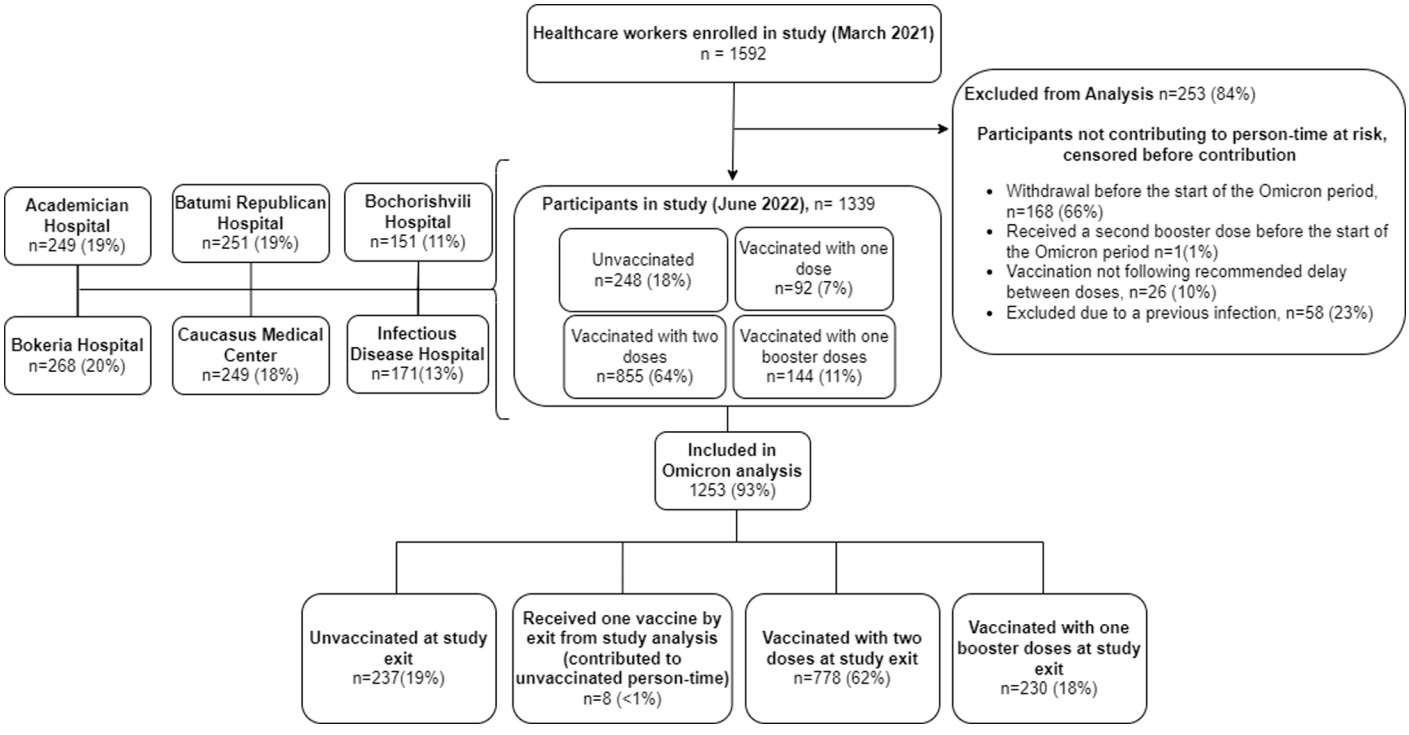

**Fig 2. Flowchart showing enrolment of healthcare workers in COVID-19 vaccine effectiveness study, Georgia, 2021–2022.**

Relative VE of a booster dose compared to PVS against symptomatic infection was 58% (95% CI 1–82) for 7 days – 29 days from first booster dose receipt, 21% (95% CI -33–54) for 30 days – 59 days from first booster dose receipt, and -9% (95%CI -82–34) for ≥ 60 days from first booster dose receipt (Table 2, Fig 4).

Relative VE for a first booster dose of BNT162b2 vaccine compared to only a PVS (≥ 150d) of BNT162b2 vaccine against symptomatic infection was 77% (95%CI 3–94) for 7 days – 29 days from the first booster dose receipt, 26% (95% CI -47–63) for 30 days – 59 days from the first booster dose receipt, and -16% (95%CI -103–34) for ≥ 60 days from the first booster dose receipt (Table 2, Fig 4). VE for other booster vaccines could not be calculated due to small sample size.

Absolute VE of a first booster dose against medically attended infection was 45% (95% CI -121–86) for 7 days – 29 days from the first booster dose receipt, 29% (95% CI -71–71) for 30 days – 59 days from the first booster dose receipt, and 21% (95% CI -104–70) for ≥ 60 days from booster dose receipt VE (Table 2, Fig 4).

Relative VE for a booster dose compared to PVS against medically attended infection was 45% (95% CI -121–87) for 7d – 29d from booster dose receipt, 33% (95% CI -67–73), for 30d – 59 days from the first booster dose receipt, and 27% (95%CI -90–72). for ≥ 60d from the first booster dose receipt (Table 2, Fig 4).

## Discussion

During Omicron circulation in early 2022 in Georgia, we found moderate COVID-19 absolute VE (≥40%) of the first booster dose against both symptomatic and medically attended infection that rapidly waned after 29 days in a cohort of HCWs with a high rate of previous infection. Relative VE against symptomatic infection was slightly higher (58%) for a first booster dose compared to PVS, but relative VE also quickly waned after 29 days since vaccination (Table 2, Fig 4).

Our finding of moderate VE against symptomatic infection that rapidly waned during Omicron is similar to findings from other studies showing moderate VE using ancestral monovalent vaccines against Omicron that decreased over time.

**Table 1. Demographic, occupational, and clinical characteristics among study participants by vaccination status at the beginning of the study period on 1 January 2022, Georgia (N = 1253).**

| | Missing | All Participants | Unvaccinated | Vaccinated with Primary Series | First Booster dose |
|---|---|---|---|---|---|
| **Age** | 0 | n = 1253 | n = 248 | n = 855 | n = 141 |
| Median (IQR) | | 41 (30-53) | 39 (28-52) | 40 (30-52) | 49 (37-60) |
| **Age group** | 0 | n = 1253 | n = 248 | n = 855 | n = 141 |
| <20, n(%) | | 13 (1) | 1 (<1) | 12 (1) | 0 (0) |
| 20-29, n(%) | | 281 (22) | 68 (27) | 193 (23) | 16 (11) |
| 30-39, n(%) | | 275 (22) | 57 (23) | 192 (22) | 26 (18) |
| 40-49, n(%) | | 268 (21) | 48 (19) | 189 (22) | 30 (21) |
| 50-59, n(%) | | 262 (21) | 51 (21) | 175 (20) | 32 (23) |
| 60+, n(%) | | 154 (12) | 23 (9) | 94 (11) | 37 (26) |
| **Sex** | 0 | n= 1253 | n= 248 | n= 855 | n= 141 |
| F, n(%) | | 1058 (84) | 206 (83) | 731 (85) | 114 (81) |
| M, n(%) | | 195 (16) | 42 (17) | 124 (15) | 27 (19) |
| **Hospital** | 0 | n= 1253 | n= 248 | n= 855 | n= 141 |
| Academician Hospital, n(%) | | 229 (18) | 27 (11) | 154 (18) | 46 (33) |
| Batumi Republican Hospital, n(%) | | 238 (19) | 31 (12) | 174 (20) | 28 (20) |
| Bochoroshvili Hospital, n(%) | | 137 (11) | 31 (12) | 96 (11) | 8 (6) |
| Bokeria Hospital, n(%) | | 249 (20) | 57 (23) | 166 (19) | 26 (18) |
| Caucasus Medical Centre, n(%) | | 236 (19) | 57 (23) | 162 (19) | 17 (12) |
| Infectious Disease Hospital, n(%) | | 164 (13) | 45 (18) | 103 (12) | 16 (11) |
| **Occupation/Role in hospital** | 0 | n= 1253 | n= 248 | n= 855 | n= 141 |
| Nurse or Midwife, n(%) | | 487 (39) | 103 (42) | 357 (42) | 24 (17) |
| Medical Doctor, n(%) | | 257 (21) | 20 (8) | 160 (19) | 76 (54) |
| Other, n(%) | | 509 (41) | 125 (50) | 338 (40) | 41 (29) |
| **Household size** | 0 | n= 1253 | n= 248 | n= 855 | n= 141 |
| 1-3, n(%) | | 574 (46) | 125 (50) | 368 (43) | 77 (55) |
| 4-5, n(%) | | 494 (39) | 93 (38) | 355 (42) | 41 (29) |
| 6+, n(%) | | 185 (15) | 30 (12) | 132 (15) | 23 (16) |
| **Number of chronic conditions** | 0 | n= 1253 | n= 248 | n= 855 | n= 141 |
| 0, n(%) | | 937 (75) | 184 (74) | 659 (77) | 86 (61) |
| 1, n(%) | | 248 (20) | 53 (21) | 154 (18) | 40 (28) |
| 2+, n(%) | | 68 (5) | 11 (4) | 42 (5) | 15 (11) |
| **BMI** | 0 | n= 1253 | n= 248 | n= 855 | n= 141 |
| Underweight or normal, n(%) | | 570 (45) | 115 (46) | 393 (46) | 56 (40) |
| Overweight, n(%) | | 386 (31) | 70 (28) | 262 (31) | 53 (38) |
| Obese, n(%) | | 297 (24) | 63 (25) | 200 (23) | 32 (23) |
| **Smoking status** | 0 | n= 1253 | n= 248 | n= 855 | n= 141 |
| Currently smokes, n(%) | | 304 (24) | 57 (23) | 204 (24) | 38 (27) |
| Never smokes, n(%) | | 835 (67) | 169 (68) | 582 (68) | 80 (57) |
| Previously smokes, n(%) | | 114 (9) | 22 (9) | 69 (8) | 23 (16) |
| **Self-assessed health status** | 0 | n= 1253 | n= 248 | n= 855 | n= 141 |
| Excellent, n(%) | | 101 (8) | 22 (9) | 65 (8) | 14 (10) |
| Very good, n(%) | | 201 (16) | 39 (16) | 136 (16) | 25 (18) |
| Good, n(%) | | 420 (34) | 84 (34) | 291 (34) | 40 (28) |
| Fair, n(%) | | 513 (41) | 94 (38) | 356 (42) | 60 (43) |

*(Continued)*

**Table 1.** (Continued)

| | Missing | All Participants | Unvacci-nated | Vaccinated with Primary Series | First Booster dose |
|---|---|---|---|---|---|
| Poor, n(%) | | 18 (1) | 9 (4) | 7 (<1) | 2 (1) |
| Hands on care | 0 | n= 1253 | n= 248 | n= 855 | n= 141 |
| No, n(%) | | 581 (46) | 134 (54) | 385 (45) | 58 (41) |
| Yes, n(%) | | 672 (54) | 114 (46) | 470 (55) | 83 (59) |
| **Face-to-face patient contact** | 0 | n= 1253 | n= 248 | n= 855 | n= 141 |
| No, n(%) | | 266 (21) | 67 (27) | 172 (20) | 24 (17) |
| Yes, n(%) | | 987 (79) | 181 (73) | 683 (80) | 117 (83) |
| **Previous COVID-19 infection before the start of the analysis period (confirmed by RT-PCR, Rapid antigen test, or serology)** | 0 | n= 1253 | n= 248 | n= 855 | n= 141 |
| No, n(%) | | 171 (14) | 35 (14) | 113 (13) | 21 (15) |
| Yes, n(%) | | 1082 (86) | 213 (86) | 742 (87) | 120 (85) |
| **Delay between PVS and start of person-time contribution** | 257 | n= 996 | n= 0 | n= 855 | n= 141 |
| Median (IQR) | | 137 (115.8-190) | – | 132 (112-159) | 235 (187-268) |
| **Delay between first booster dose and start of person-time contribution** | 1112 | n= 141 | n= 0 | n= 0 | n= 141 |
| Median (IQR) | | 31 (14-50) | – | – | 31 (14-50) |
| **PVS product at start of person-time contribution** | 5 | n= 1248 | n= 248 | n= 850 | n= 141 |
| AstraZeneca, n(%) | | 23 (2) | 0 (0) | 12 (1) | 11 (8) |
| BNT162b, n(%) | | 678 (54) | 0 (0) | 581 (68) | 97 (69) |
| Sinopharm, n(%) | | 224 (18) | 0 (0) | 204 (24) | 20 (14) |
| Sinovac, n(%) | | 52 (4) | 0 (0) | 42 (5) | 10 (7) |
| Unvaccinated, n(%) | | 248 (20) | 248 (100) | 0 (0) | 0 (0) |
| **First booster dose product at start of person-time contribution** | 1112 | n= 141 | n= 0 | n= 0 | n= 141 |
| BNT162b, n(%) | | 130 (92) | 0 (0) | 0 (0) | 130 (92) |
| Sinopharm, n(%) | | 11 (8) | 0 (0) | 0 (0) | 11 (8) |

A systematic review of VE against symptomatic infection during the Omicron period found pooled estimates of VE after any PVS and any booster dose waned over time. VE after 1 month from PVS was 53%, after 6 months was 14%, and decreased to 9% after 9 months since PVS. VE after 1 month since receiving any booster dose was 60%, after 6 months VE was 23%, and after 9 months VE was 13% [18]. A multi-country study in Europe of COVID-19 VE against symptomatic infection during Omicron (Dec 2021 – June 2022) also found significant waning in protection. All-product PVS VE was 60% < 90 days since vaccination and 29% ≥ 180 after vaccination. All-product first booster VE was 56% < 90 days since vaccination and 3% ≥ 180 after vaccination [19].

A study of COVID-19 VE against symptomatic infection during Omicron from England (November 2021 to January 2022) showed that booster dose BNT162b2 VE against symptomatic infection was 67% following 1 week of booster dose receipt and then receded to 46% after 10 weeks [17]. Another study in England of COVID-19 VE against symptomatic infection during BA.1 and BA.2 (January – March, 2022) found VE of a first booster dose against symptomatic infection with BA.1 or BA.2 1 week after vaccination was 71% and 74%, respectively. VE against the same outcome >15 weeks after vaccination for BA.1 and BA.2 was 37% and 44%, respectively [20]. VE of PVS against symptomatic infection from <2 weeks since vaccination for BA.1 and BA.2 was 63% and 64%, respectively. VE of PVS against symptomatic infection from 25 + weeks since vaccination for BA.1 and BA.2 was 15% and 28%, respectively [20]. Another study of COVID-19 VE

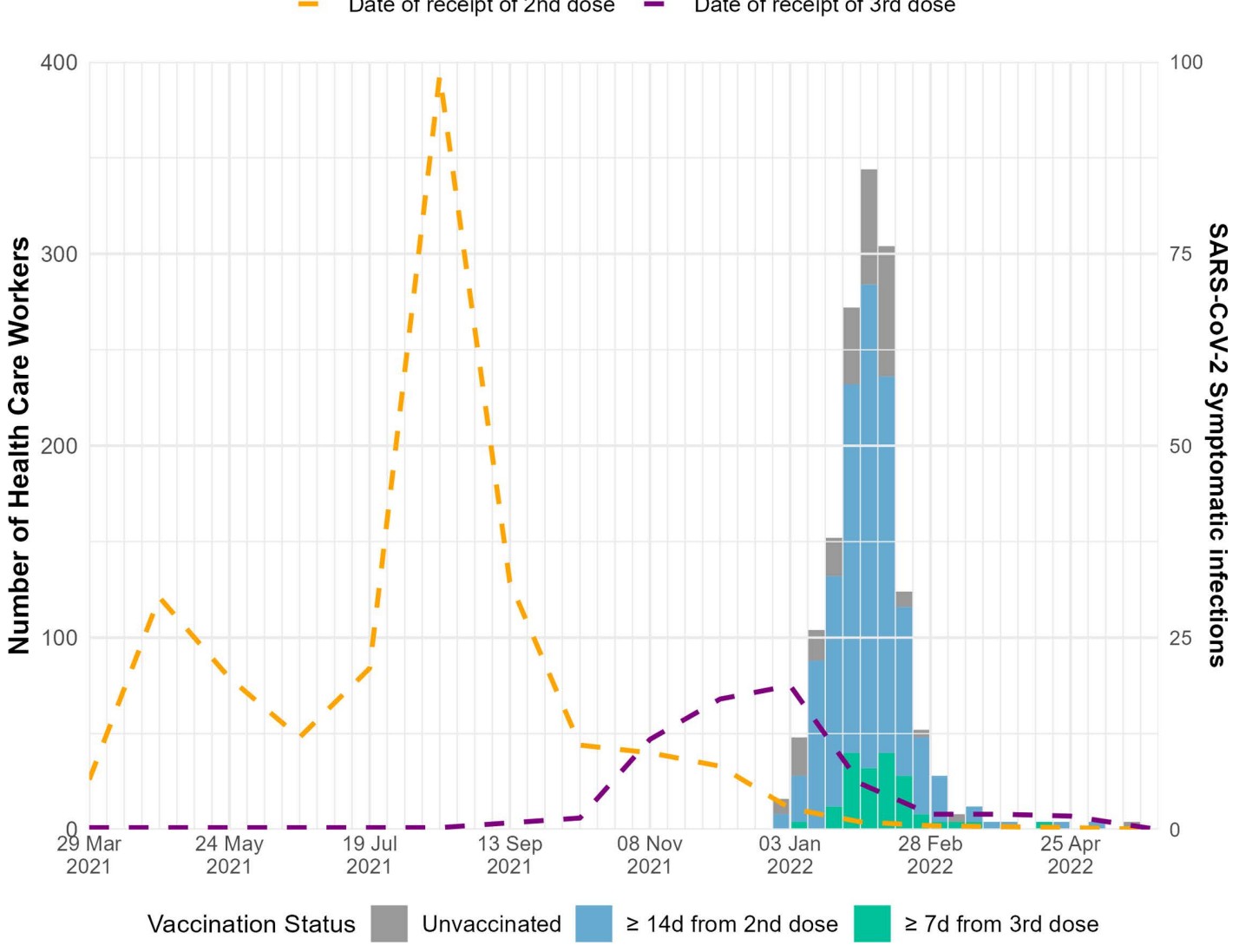

**Fig 3. Number of symptomatic SARS-CoV-2 positive healthcare workers by vaccination status and time since vaccination, January - June 2022.**

against symptomatic infection during Omicron from Canada (December 2021) showed that VE of PVS against symptomatic infection was 36% from 7–59 days since vaccination and 15% from 120–179 days since vaccination [21].

The above studies indicate overall a slower decline in VE compared to our study, which found a rapid decline in VE. Our HCW setting differs from these studies, all of which are in the general population. HCW are likely to have higher exposure levels to SARS-CoV-2 than the general population [15]. Within our study population, the epidemic curve was extremely steep (Fig 3); 95% of HCWs infected during the study period had a symptomatic SARS-CoV-2 infection between 1 January and end of February. This comprises 28% of the total study population. This force of infection, and the context of COVID-19 vaccine being at least a partially leaky vaccine [22,23], meaning that protection may be affected by a number of exposures, may explain at least in part the more rapid decline in VE observed in our study.

The higher point estimates for relative VE compared to absolute VE suggest that the risk of symptomatic infection among individuals who completed their primary series more than 150 days ago was higher than the risk of symptomatic

Table 2. Absolute and Relative Vaccine Effectiveness of primary vaccination series (PVS) vs first booster dose in protecting against symptomatic PCR and/or RAT-confirmed SARS-CoV-2 infection, overall and for BNT162b2 only, and against medically attended COVID-19, for the Omicron-predominant period only, Georgia, January 1 – June 1, 2022.

| | N participants | Total person-time (days) | PCR-confirmed symptomatic COVID-19 infection | RAT-confirmed symptomatic COVID-19 infection | All symptomatic COVID-19 infections | Unadjusted VE or rVE | (95% CI) | Absolute VE or Relative VE* | (95% CI) |
|---|---|---|---|---|---|---|---|---|---|
| Absolute vaccine effectiveness (all vaccines) | 435 | | | | | | | | |
| **Time since first booster vs. unvaccinated** | | | | | | | | | |
| Unvaccinated [ref] | 248 | 23806 | 38 | 25 | 63 | | | | |
| 7d – 29d from first booster dose | 120 | 2178 | 4 | 2 | 6 | 34.0 | (-63; 73.3) | 40.4 | (-55.9; 77.2) |
| 30d-59d from booster dose | 150 | 3655 | 13 | 2 | 15 | -35.1 | (-143; 24.9) | -8.6 | (-103.9; 42.1) |
| ≥60d from first booster dose | 152 | 11497 | 11 | 7 | 18 | -53.8 | (-158.7; 8.6) | -45.9 | (-155.7; 16.7) |
| Relative vaccine effectiveness (all vaccines) | 1070 | | | | | | | | |
| **Time since first booster doses vs. PVS (All vaccines)** | | | | | | | | | |
| 150d+ from PVS [ref] | 698 | 51752 | 107 | 41 | 148 | | | | |
| 7d – 29d from first booster dose | 120 | 2178 | 4 | 2 | 6 | 58.6 | (4.6; 82.1) | 57.5 | (0.5; 81.9) |
| 30d-59d from first booster dose | 150 | 3655 | 13 | 2 | 15 | 18 | (-39.6; 51.8) | 21.4 | (-32.9; 53.6) |
| ≥60d from first booster dose | 152 | 11497 | 11 | 7 | 18 | -2.6 | (-68.8; 37.6) | -9.4 | (-82; 34.3) |
| Relative vaccine effectiveness of BNT162b2 | 537 | | | | | | | | |
| **Time since first booster doses vs. PVS (BNT162b2)** | | | | | | | | | |
| 150d+ from PVS [ref] | 451 | 32530 | 70 | 30 | 100 | | | | |
| 7d – 29d from first booster dose | 66 | 1255 | 1 | 1 | 2 | 77.9 | (6.6; 94.8) | 77.4 | (3.4; 94.7) |
| 30d-59d from first booster dose | 89 | 2114 | 6 | 2 | 8 | 26.2 | (-51.2; 63.9) | 26.4 | (-47.2; 63.2) |

(Continued)

**Table 2.** (Continued)

| | N participants | Total person-time (days) | PCR-confirmed symptomatic COVID-19 infection | RAT-confirmed symptomatic COVID-19 infection | All symptomatic COVID-19 infections | Unadjusted VE or rVE | (95% CI) | Absolute VE or Relative VE* | (95% CI) |
|---|---|---|---|---|---|---|---|---|---|
| ≥60d from first booster dose | 100 | 7455 | 10 | 5 | 15 | -3.7 | (-78.8; 39.9) | -15.6 | (-103.1; 34.2) |
| Absolute vaccine effectiveness against Medically attended COVID-19 (all vaccines) | | | | | | | | | |
| **Time since first booster dose vs. unvaccinated** | 435 | | | | | | | | |
| Unvaccinated [ref] | 248 | 23806 | 24 | 12 | 36 | | | | |
| 7d – 29d from first booster dose | 120 | 2178 | 1 | 2 | 3 | 42.4 | (-109.6; 84.2) | 45.3 | (-120.8; 86.4) |
| 30d-59d from first booster dose | 150 | 3655 | 5 | 0 | 5 | 6.3 | (-145.4; 64.2) | 29.1 | (-71; 70.6) |
| ≥60d from first booster dose | 152 | 11497 | 3 | 2 | 5 | 15.2 | (-110.3; 65.8) | 21.2 | (-103.9; 69.5) |
| Absolute vaccine effectiveness against Medically attended COVID-19 (all vaccines) | | | | | | | | | |
| **Time since first booster doses vs. PVS** | 822 | | | | | | | | |
| 150d+ from PVS [ref] | 698 | 51752 | 51 | 22 | 73 | | | | |
| 7d – 29d from first booster dose | 120 | 2178 | 1 | 2 | 3 | 58.2 | (-40.3; 87.5) | – | – |
| 30d-60d from first booster dose | 150 | 3655 | 5 | 0 | 5 | 40.8 | (-47.3; 76.2) | 32.8 | (-66.5; 72.8) |
| ≥60d from first booster dose | 152 | 11497 | 3 | 2 | 5 | 37.2 | (-56.4; 74.8) | 26.6 | (-90.1; 71.7) |

*VE estimates were adjusted for hospital site (strata) age groups, sex, any underlying chronic condition, self-assessed health status, household size, daily face-to-face patient contact, previous COVID-19 infection, and smoking status.

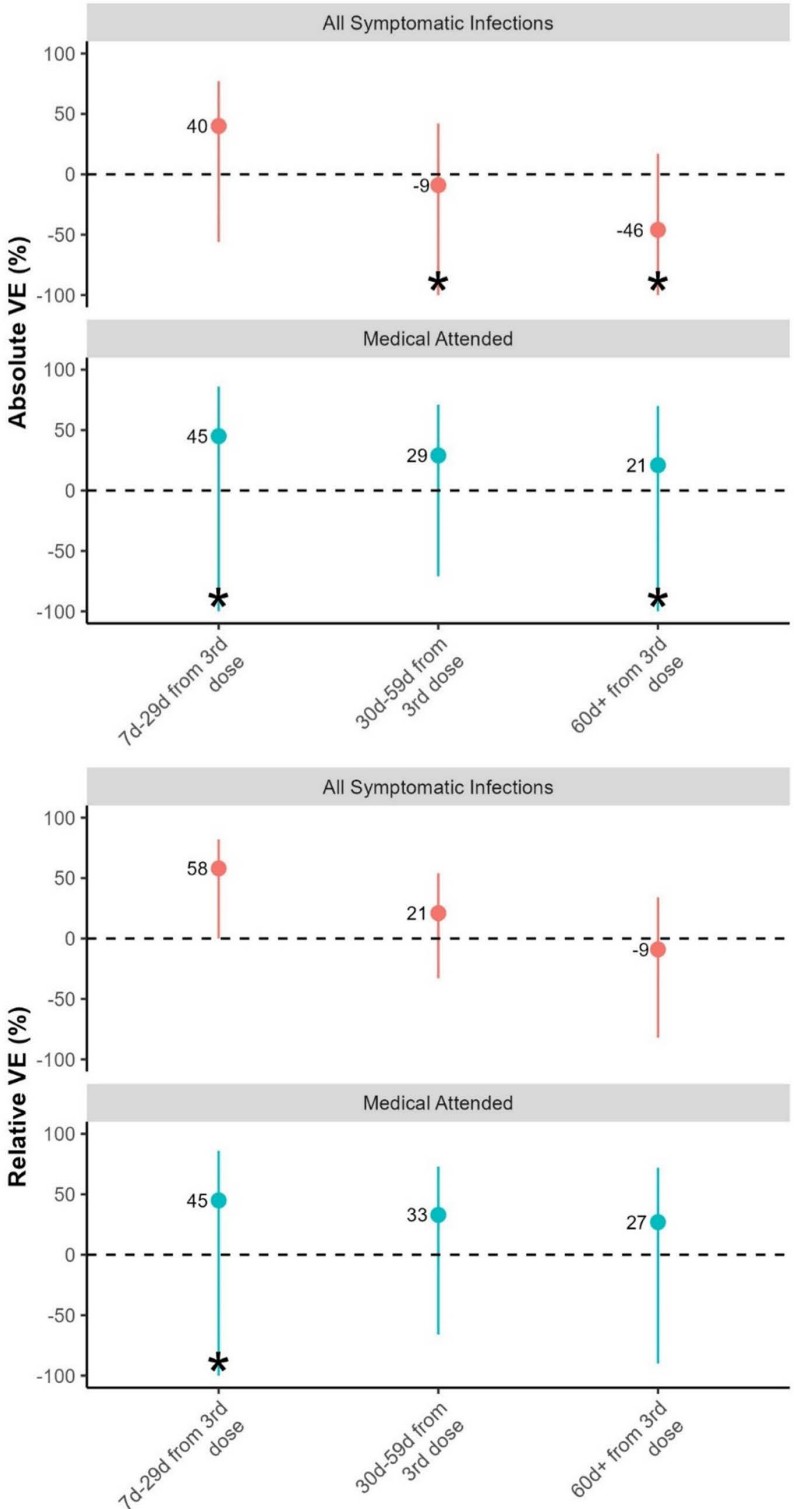

**Fig 4. Absolute and relative COVID-19 vaccine effectiveness against symptomatic infection and medically attended HCWs across different time periods, for Omicron-predominant period, Georgia, 2022.**

infection among the unvaccinated. We would expect that the risk of symptomatic infection among those vaccinated with primary series more than 150 days ago would be the same or if anything slightly lower than the unvaccinated, in case of residual protection. This difference may reflect differences between unvaccinated HCWs and those vaccinated more than 150 days ago in terms of virus exposure or variability in seroprotection related to previous infection. However, as confidence intervals overlap between estimates, there is uncertainty in this difference, and it may be due to random error.

While we found that moderate VE against symptomatic infection rapidly waned, we were not able to evaluate VE against more severe endpoints than medically attended COVID-19. Studies have found that the COVID-19 vaccine continues to be effective in preventing severe disease. A multi-country study in Europe during a period of Omicron-predominant circulation (December 2021 – July 2022) found that COVID-19 VE against hospitalization was 43% for PVS and 59% for a first booster dose for ≥ 150 days between the last PVS and symptom onset or last PVS and booster dose with 85% VE if illness onset was 14–59 days after booster receipt [24]. A study in the Netherlands from October – December 2023 looking at VE in those aged ≥60 during found VE against hospitalization was 71% [25].

Our study had several strengths. We used PCR-confirmed infection and serology to determine participants with prior infections. We used both PCR testing and rapid antigen tests to identify infections more completely among study participants. We were able to adjust for different confounders in our VE analysis. We were also able to evaluate VE against the more severe endpoint of medically attended infections.

Our study also had limitations. Our study was underpowered to detect the true effect in some sub-group analyses, particularly those with a low number of events. Therefore, the results of sub-group analyses should be interpreted cautiously. Our study was not powered to assess VE against more severe outcomes such as hospitalization, ICU admission, or death. Our sample size was low for estimating certain time periods of VE and medically attended VE, which resulted in wide CIs. We did not account for variability in routine testing of asymptomatic participants. However, we focused our analysis on symptomatic infection, an outcome which is less likely to be affected by the variability in routine asymptomatic testing. Because we could not use serology tests to determine previous infection status among the 22% of participants who had received inactivated vaccines, we likely underestimated the number of previous infections in this population. However, because we did not conduct a stratified analysis based on previous infection, this underestimation should not have biased our VE estimates. We did not have the resources to do micro-neutralization tests which would have given us more accurate results. We may have missed previous infections not captured by PCR testing or serology testing. While we could not find published data on the sensitivity of the ECLIA serology test against Omicron subvariants, the pooled sensitivity of the test against the original strain for IgG and IgM were 0.93 and 0.85, respectively, compared to rRT-PCR [26]. We were unable to stratify VE by prior infection status or estimate vaccine products other than BNT162b2 due to the small sample size. The high rates of prior infections among participants likely led to an underlying level of immunity not seen in uninfected populations. While the immunity likely blunted the magnitude of VE, it reflects the real world setting of Georgia at the time of the study. HCWs in our study enrolled voluntarily, which may have caused selection bias. Participants may have been healthier and more likely to be compliant with the study criteria compared to HCWs in the hospital. However, we do not know if this would have an overall impact on VE. While the hospitals included in our study are large hospitals in the Georgian capital, we do not have demographic and comorbidity data for all Georgian HCWs, and therefore we are unable to determine the extent to which our study population is representative of all HCWs in the country. However, we managed to enroll 40% of HCWs in the 6 study hospitals [10]. Enrolling a large proportion of HCWs helps generalize our findings to all HCWs in Georgia.

Overall, we found that COVID-19 booster doses were moderately effective at preventing symptomatic infection in hospital based HCWs in Georgia but VE rapidly waned. Our overall VE results should be interpreted cautiously as we used both inactivated and mRNA vaccines in our study. This study was conducted in the context of high previous infection and a high force of infection. These results suggest that timing COVID-19 vaccination campaigns prior to periods of anticipated high SARS-CoV-2 incidence, if practical, could be important to reduce COVID-19 infections. The moderate but waning VE

in this study against symptomatic and medically attended infection combined with the more durable COVID-19 VE against severe outcomes described in other studies, underscores the need for broader efforts in Georgia to increase vaccine coverage among priority groups such as HCWs, and to encourage the adoption of guidelines that encourage high-risk individuals to receive annual COVID-19 revaccination, as recommended by WHO [27].

## Acknowledgments

We are grateful to the study team at the Georgia National Centers for Disease Control and at all the participating study hospitals. In addition, we would like to thank the following colleagues: Tamuna Zardiashvili, C Jason McKnight, and Irina Begiashvilli (WHO/Georgia), Pernille Jorgensen and Diogo Simao Lemos (WHO/Europe), Lindsey Duca (US CDC), Marta Valenciano and Alain Moren (Epiconcept). This study has been conducted within the framework of the WHO Strategic Preparedness and Response plan and WHO/European Regional Office Response for COVID-19.

## Author contributions

**Conceptualization:** Madelyn Yiseth Rojas Castro, Giorgi Chakhunashvili, Richard Pebody, Esther Kissling, Mark A Katz, Lia Sanodze.

**Data curation:** Caleb L Ward, Madelyn Yiseth Rojas Castro, Giorgi Chakhunashvili, Iris Finci, Esther Kissling, Lia Sanodze.

**Formal analysis:** Madelyn Yiseth Rojas Castro, Esther Kissling.

**Funding acquisition:** Caleb L Ward, Richard Pebody, Mark A Katz, Lia Sanodze.

**Investigation:** Giorgi Chakhunashvili, Nazibrola Chitadze, Mark A Katz, Lia Sanodze.

**Methodology:** Madelyn Yiseth Rojas Castro, Giorgi Chakhunashvili, Nazibrola Chitadze, Iris Finci, Richard Pebody, Esther Kissling, Mark A Katz, Lia Sanodze.

**Project administration:** Caleb L Ward, Giorgi Chakhunashvili, Nazibrola Chitadze, Iris Finci, Mark A Katz, Lia Sanodze.

**Resources:** Mark A Katz, Lia Sanodze.

**Software:** Caleb L Ward, Madelyn Yiseth Rojas Castro, Iris Finci, Esther Kissling.

**Supervision:** Caleb L Ward, Giorgi Chakhunashvili, Iris Finci, Richard Pebody, Esther Kissling, Mark A Katz, Lia Sanodze.

**Validation:** Caleb L Ward, Madelyn Yiseth Rojas Castro, Giorgi Chakhunashvili, Iris Finci, Esther Kissling, Lia Sanodze.

**Visualization:** Caleb L Ward, Madelyn Yiseth Rojas Castro, Giorgi Chakhunashvili, Iris Finci, Mark A Katz.

**Writing – original draft:** Caleb L Ward.

**Writing – review & editing:** Caleb L Ward, Madelyn Yiseth Rojas Castro, Giorgi Chakhunashvili, Nazibrola Chitadze, Iris Finci, Richard Pebody, Esther Kissling, Mark A Katz, Lia Sanodze.

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
