## [Decision Letter · Decision Letter 0]

29 Jan 2025

PONE-D-24-40712COVID-19 Vaccine effectiveness among Healthcare Workers during the Omicron Period in the country of Georgia, January – June 2022PLOS ONE

Dear Dr. Ward,

Thank you for submitting your manuscript to PLOS ONE. After careful consideration, we feel that it has merit but does not fully meet PLOS ONE’s publication criteria as it currently stands. Therefore, we invite you to submit a revised version of the manuscript that addresses the points raised during the review process. Please submit your revised manuscript by Mar 15 2025 11:59PM. If you will need more time than this to complete your revisions, please reply to this message or contact the journal office at plosone@plos.org . Please include the following items when submitting your revised manuscript:

We look forward to receiving your revised manuscript.

Kind regards,

Eleonora Nicolai, PhD

Academic Editor

PLOS ONE

Journal Requirements:

“This study was supported by funding from the World Health Organization/European Regional Office, the US

Centers for Disease Control and Prevention (Cooperative Agreement No.: NU2GGH002093), and the Public Health

Institute”

Reviewers' comments:

Reviewer's Responses to Questions

**Comments to the Author**

1. Is the manuscript technically sound, and do the data support the conclusions?

Reviewer #1: Yes

Reviewer #2: Yes

2. Has the statistical analysis been performed appropriately and rigorously? 

Reviewer #1: Yes

Reviewer #2: Yes

3. Have the authors made all data underlying the findings in their manuscript fully available?

Reviewer #1: Yes

Reviewer #2: No

4. Is the manuscript presented in an intelligible fashion and written in standard English?

Reviewer #1: Yes

Reviewer #2: Yes

5. Review Comments to the Author

Reviewer #1: Abstract:

• Lines 12–15 (Abstract): The study presents original data on vaccine effectiveness (VE) among healthcare workers (HCWs) in Georgia during the Omicron period, a topic of significant public health interest. It appears that results have not been published elsewhere, but authors should explicitly confirm this in the manuscript.

Lines 1–37 (Abstract): The language is clear, but sentences such as “Absolute VE for a first booster was 40%… at ≥ 60 days” (Lines 28–30) could be simplified for readability.

Methodology and Analysis :

• Lines 61–116 (Methods): The cohort design and use of Cox proportional hazards models are appropriate. However:

• Provide more details on how inconsistent testing among HCWs (Line 73) was accounted for in VE estimation.

• Clarify why serology tests for inactivated vaccines could not determine prior infections (Line 97). Could this limitation bias results?

• Adjustments for confounders (Lines 106–109) are comprehensive, but their selection rationale should be better justified.

• (lines:76-78)Serology samples were tested for anti-nucleocapsid antibodies and anti-spike antibodies by Roche Elecsys Anti-SARS-CoV-2 S immunoassay electrochemiluminescence immunoassay (ECLIA): what is the variant strain which was used as coated antigen in kits ? and what is the sensitivity of kit for detection the Ab against predominant circulating variants ?

• vaccine effectiveness (lines:80-99) why did not you use VNT to detect the vaccine antibody titer in vaccinated HCWs against predominant circulating variants?

Conclusions and Data Support :

• Lines 163–234 (Discussion):

• The conclusion of “moderate but waning VE” is supported by the data. However, the authors should elaborate on the implications of waning VE for booster policy in middle-income countries (Lines 231–234).

• Lines 163–234 :The writing is coherent, but the overuse of technical terms may limit accessibility to a broader audience. Simplify where possible.

• Discuss whether high prior infection rates (Line 122) could have influenced observed VE.

General Recommendations:

1. Figures and Tables: The figures (e.g., VE over time) are informative but need better integration into the discussion to illustrate key findings.

2. Limitations (Lines 221–227): Expand on the potential selection bias due to voluntary participation and address whether the sample is representative of HCWs in Georgia.

3. Implications for Future Research: Suggest areas for further study, such as VE against severe outcomes.

Reviewer #2: he study report on COVID-19 vaccine efficacy (VE) against symptomatic infection from January 2022 to June 2022, during the Omicron era in Georgia. Participants were selected from healthcare workers vaccinated with either mRNA or inactivated vaccines available. The report indicates that the booster dose is moderately effective in preventing symptomatic infection in healthcare workers (HCWs) in Georgia, but its effectiveness wanes over time. These findings have been previously reported in other countries with similar conclusions, and the same authors had published a similar study (https://doi.org/10.1371/journal.pone.0307805). In this study, the authors determined the VE of COVID-19 vaccines containing the ancestral virus during Omicron circulation. The authors should discuss other studies that reported the VE of ancestral monovalent vaccines against Omicron. Furthermore, some participants received inactivated vaccines, while others received mRNA vaccines. The results should be interpreted cautiously since both vaccines are produced differently. Therefore, it is not clear whether these findings and conclusions represent the VE of mRNA or inactivated vaccines. Authors should comment and discuss on this in the discussion.

Minor edit: The Sinopharm vaccine is written as BBIBP-CorV in some places, but used differently as BIBP-CorV in other places. Please correct.

6. PLOS authors have the option to publish the peer review history of their article (what does this mean? ). If published, this will include your full peer review and any attached files.

**Do you want your identity to be public for this peer review?** For information about this choice, including consent withdrawal, please see our Privacy Policy .

Reviewer #1: **Yes: ** mohamed samy abousenna

Reviewer #2: No

---

## [Author Response · Author response to Decision Letter 1]

31 Mar 2025

14 March 2025

Dear Plos One Editors,

Thank you and thanks to the reviewers for the thorough review of our manuscript. We have done our best to address all of the questions raised by reviewers in the point-by-point responses below.

Thanks again for considering our manuscript for publication.

Sincerely,

Caleb Ward

We have changed the manuscript and file names to match PLOS ONE’s style requirements.

2-3. We note that the grant information you provided in the ‘Funding Information’ and ‘Financial Disclosure’ sections do not match.

We have changed the funding information section to reflect the appropriate funders. The funders had no role in study design, data collection and analysis, decision to publish, or preparation of the manuscript.

4. Data availability

Data contain potentially identifiable and sensitive participant information and cannot be shared. The data availability statement has been updated.

5. Ethics statement

The ethics statement has been added to the methods section of the manuscript.

6. Please include captions for your Supporting Information files at the end of your manuscript

The following caption was added at the end of the manuscript.

Supporting Information. Ethics Approvals from NCDC and WHO.

7. Please review your reference list to ensure that it is complete and correct.

The reference list is complete and correct.

Reviewer #1: Abstract:

• Lines 12–15 (Abstract): The study presents original data on vaccine effectiveness (VE) among healthcare workers (HCWs) in Georgia during the Omicron period, a topic of significant public health interest. It appears that results have not been published elsewhere, but authors should explicitly confirm this in the manuscript.

We have added the line “but VE estimates against the Omicron variant have not been previously published” in the introduction.

Lines 1–37 (Abstract): The language is clear, but sentences such as “Absolute VE for a first booster was 40%… at ≥ 60 days” (Lines 28–30) could be simplified for readability.

We added “following vaccination” to the abstract to clarify the results so the sentence now reads:

Absolute VE for a first booster was 40% (95% Confidence Interval (CI) -56 – 77) at 7– 29 days following vaccination, -9% (95% CI -104 – 42) at 30 – 59 days following vaccination, and -46% (95% CI -156 – 17) at ≥ 60 days following vaccination.

Methodology and Analysis :

• Lines 61–116 (Methods): The cohort design and use of Cox proportional hazards models are appropriate. However:

• Provide more details on how inconsistent testing among HCWs (Line 73) was accounted for in VE estimation.

We did not account for variability in routine testing of asymptomatic participants. However, we focused our analysis on symptomatic infection, and outcome which is less likely to be affected by the variability in routine asymptomatic testing.

We have added the above text to the limitations section.

• Clarify why serology tests for inactivated vaccines could not determine prior infections (Line 97). Could this limitation bias results?

We could not use serology test results to determine prior infection for participants who were previously vaccinated with an inactivated vaccine (BBIBP-CorV or CoronaVac), as these vaccines routinely generate anti-nucleocapsid antibodies, and therefore positive serology test results in these participants could reflect either prior infection or prior vaccination.

We have added the additional language above to the methods section.

Because we could not use serology tests to determine previous infection status among the 22% of participants who had received inactivated vaccines, we likely underestimated the number of previous infections in this population. However, because we did not conduct stratified analysis based on previous infection, this underestimation should not have biased our VE estimates.

We have added the text above to the limitations section.

• Adjustments for confounders (Lines 106–109) are comprehensive, but their selection rationale should be better justified.

We have added the following sentence to the methods section: “These variables were chosen because they were previously found to be associated with vaccine uptake and effectiveness.”

• (lines:76-78)Serology samples were tested for anti-nucleocapsid antibodies and anti-spike antibodies by Roche Elecsys Anti-SARS-CoV-2 S immunoassay electrochemiluminescence immunoassay (ECLIA): what is the variant strain which was used as coated antigen in kits ? and what is the sensitivity of kit for detection the Ab against predominant circulating variants ?

Added to the limitations.

The ECLIA assay used antigens from the original SARS-CoV-2 strain.

We did not find any data on the sensitivity of these tests for antibodies from Omicron ba1 and ba2. We used antibody kits that detected the original strain and results may differ from the predominant strain in our study.

We could not find any published studies on PubMed or pre-print servers that evaluated the sensitivity of the ECLIA test against Omicron subvariants. However, one study that evaluated the sensititivity of ECLIA tests against the original strain found that the sensitivity of the test for IgG and IgM were 0.93 and 0.85.1

Because of this we added the following sentence to the limitations,

“We may have missed previous infections not captured by PCR testing or serology testing. While we could not find published data on the sensitivity of the ECLIA serology test against Omicron subvariants, the pooled sensitivity of the test against the original strain for IgG and IgM were 0.93 and 0.85, respectively, compared to rRT-PCR.”

1 Vengesai, A., Midzi, H., Kasambala, M. et al. A systematic and meta-analysis review on the diagnostic accuracy of antibodies in the serological diagnosis of COVID-19. Syst Rev 10, 155 (2021). https://doi.org/10.1186/s13643-021-01689-3

• vaccine effectiveness (lines:80-99) why did not you use VNT to detect the vaccine antibody titer in vaccinated HCWs against predominant circulating variants?

We did not have the resources to conduct micro-neutralization assays to assess SARS-CoV-2 infection.

We have added this sentence to the limitations section.

Conclusions and Data Support :

• Lines 163–234 (Discussion):

• The conclusion of “moderate but waning VE” is supported by the data. However, the authors should elaborate on the implications of waning VE for booster policy in middle-income countries (Lines 231–234).

We added the following sentence to the conclusion:

The moderate but waning VE in this study against symptomatic and medically attended infection, combined with the more durable COVID-19 VE against severe outcomes described in other studies, underscores the need for broader efforts in Georgia to increase vaccine coverage among priority groups such as HCWs, and to encourage the adoption of guidelines that encourage high-risk individuals to receive annual COVID-19 revaccination, as recommended by WHO.

• Lines 163–234 :The writing is coherent, but the overuse of technical terms may limit accessibility to a broader audience. Simplify where possible.

We have attempted to simplify the language and limit the use of technical terms where we could.

• Discuss whether high prior infection rates (Line 122) could have influenced observed VE.

We added the following sentence to the limitations, “The high rates of prior infections among participants likely led to an underlying level of immunity not seen in uninfected populations. While the immunity likely blunted the magnitude of VE, it reflects the real world setting of Georgia at the time of the study.”

General Recommendations:

1. Figures and Tables: The figures (e.g., VE over time) are informative but need better integration into the discussion to illustrate key findings.

We have integrated the tables and figures more into the discussion and referenced them in the text where relevant.

2. Limitations (Lines 221–227): Expand on the potential selection bias due to voluntary participation and address whether the sample is representative of HCWs in Georgia.

We added the following to the limitations, “Participants may have been healthier and more likely to be compliant with the study criteria compared to HCWs in the hospital. However, we do not know if this would have an overall impact on VE. While the hospitals included in our study are large hospitals in the Georgian capital, we do not have demographic and comorbidity data for all Georgian HCWs, and therefore we are unable to determine the extent to which our study population is representative of all HCWs in the country.”

3. Implications for Future Research: Suggest areas for further study, such as VE against severe outcomes.

Reviewer #2: he study report on COVID-19 vaccine efficacy (VE) against symptomatic infection from January 2022 to June 2022, during the Omicron era in Georgia. Participants were selected from healthcare workers vaccinated with either mRNA or inactivated vaccines available. The report indicates that the booster dose is moderately effective in preventing symptomatic infection in healthcare workers (HCWs) in Georgia, but its effectiveness wanes over time. These findings have been previously reported in other countries with similar conclusions, and the same authors had published a similar study (https://doi.org/10.1371/journal.pone.0307805). In this study, the authors determined the VE of COVID-19 vaccines containing the ancestral virus during Omicron circulation. The authors should discuss other studies that reported the VE of ancestral monovalent vaccines against Omicron. Furthermore, some participants received inactivated vaccines, while others received mRNA vaccines. The results should be interpreted cautiously since both vaccines are produced differently. Therefore, it is not clear whether these findings and conclusions represent the VE of mRNA or inactivated vaccines. Authors should comment and discuss on this in the discussion.

We added the following sentence to the discussion, “Our finding of moderate VE against symptomatic infection that rapidly waned during Omicron is similar to findings from other studies showing moderate VE using ancestral monovalent vaccines against Omicron that decreased over time.” 1,2

1. Andrews N, Stowe J, Kirsebom F, Toffa S, Rickeard T, Gallagher E, et al. Covid-19 Vaccine Effectiveness against the Omicron (B.1.1.529) Variant. New England Journal of Medicine. 2022 Apr 21;386(16):1532–46.

2. Kirsebom FCM, Andrews N, Stowe J, Toffa S, Sachdeva R, Gallagher E, et al. COVID-19 vaccine effectiveness against the omicron (BA.2) variant in England. The Lancet Infectious Diseases. 2022 Jul 1;22(7):931–3.

We added the following sentence to the discussion, “Our overall VE results should be interpreted cautiously as we used both inactivated and mRNA vaccines in our study.”

Minor edit: The Sinopharm vaccine is written as BBIBP-CorV in some places, but used differently as BIBP-CorV in other places. Please correct.

We corrected BIBP-CorV to BBIRP-CorV and the term is consistent throughout the manuscript.

---

## [Decision Letter · Decision Letter 1]

15 Apr 2025

COVID-19 Vaccine effectiveness among Healthcare Workers during the Omicron Period in the country of Georgia, January – June 2022

PONE-D-24-40712R1

Dear Dr. Caleb L Ward,

We’re pleased to inform you that your manuscript has been judged scientifically suitable for publication and will be formally accepted for publication once it meets all outstanding technical requirements.

Kind regards,

Eleonora Nicolai, PhD

Academic Editor

PLOS ONE

Additional Editor Comments (optional):

Dear authors, thank you for improving your manuscript according to reviewers' suggestions.

Reviewers' comments:

Reviewer's Responses to Questions

**Comments to the Author**

1. If the authors have adequately addressed your comments raised in a previous round of review and you feel that this manuscript is now acceptable for publication, you may indicate that here to bypass the “Comments to the Author” section, enter your conflict of interest statement in the “Confidential to Editor” section, and submit your "Accept" recommendation.

Reviewer #1: All comments have been addressed

2. Is the manuscript technically sound, and do the data support the conclusions?

Reviewer #1: Yes

3. Has the statistical analysis been performed appropriately and rigorously? 

Reviewer #1: Yes

4. Have the authors made all data underlying the findings in their manuscript fully available?

Reviewer #1: Yes

5. Is the manuscript presented in an intelligible fashion and written in standard English?

Reviewer #1: Yes

6. Review Comments to the Author

Reviewer #1: General Assessment

This manuscript presents a well-designed cohort study on COVID-19 vaccine effectiveness (VE) among healthcare workers (HCWs) in Georgia during the Omicron wave. It addresses a critical public health question in a middle-income country context and fills a regional data gap. The study is timely, methodologically sound, and generally well-written, with appropriate use of statistical tools such as Cox proportional hazards modeling.

1. Waning Immunity Interpretation

• The finding of VE rapidly waning within 60 days post-booster dose is striking and may suggest either a true biological decline or potential bias (e.g., higher exposure in boosted HCWs).

• While the manuscript discusses this, a more detailed comparison with immunogenicity studies or observational findings from similar HCW cohorts (e.g., Andrews et al., 2022) would help contextualize the rapid waning.

2. Combining Vaccine Platforms

• The study combines mRNA and inactivated vaccines (e.g., BNT162b2 and BBIBP-CorV) in the VE estimates, but these have distinct immunogenic profiles.

• The authors do provide stratified estimates for BNT162b2, which is commendable. However, additional emphasis on this heterogeneity and implications for interpretation should be added to the discussion.

3. Selection Bias and Generalizability

• The discussion notes potential selection bias due to voluntary participation, but does not provide a comparison to national-level HCW demographics. If such data is unavailable, state this explicitly.

• Consider expanding the limitations to clarify how this might influence VE estimates, especially if HCWs in the study were more health-conscious or had differential testing behaviors.

4.Clarity of VE Estimates in Abstract

• The abstract includes negative VE estimates, which may confuse non-expert readers. Consider clarifying that these negative values reflect non-significant findings with wide confidence intervals due to small sample sizes or waning protection.

5. Terminology Consistency

• There was previously an inconsistency in naming BBIBP-CorV as BIBP-CorV. This appears to have been corrected, but a final proofread is advised.

6. Figure Integration

• Figures and tables are informative but are not consistently discussed in the text. Specific references to Fig 4 and Table 2 within the results and discussion would improve clarity and reinforce interpretation.

7. PLOS authors have the option to publish the peer review history of their article (what does this mean? ). If published, this will include your full peer review and any attached files.

**Do you want your identity to be public for this peer review?** For information about this choice, including consent withdrawal, please see our Privacy Policy .

Reviewer #1: **Yes: ** Mohamed Samy Abousenna

---

## [Editor Report · Acceptance letter]

PONE-D-24-40712R1

PLOS ONE

Dear Dr. Ward,

I'm pleased to inform you that your manuscript has been deemed suitable for publication in PLOS ONE. Congratulations! Your manuscript is now being handed over to our production team.

Kind regards,

on behalf of

Dr. PLOS Manuscript Reassignment

Staff Editor

PLOS ONE